# CeO_2_-Zn Nanocomposite Induced Superoxide, Autophagy and a Non-Apoptotic Mode of Cell Death in Human Umbilical-Vein-Derived Endothelial (HUVE) Cells

**DOI:** 10.3390/toxics10050250

**Published:** 2022-05-16

**Authors:** Mohd Javed Akhtar, Maqusood Ahamed, Hisham Alhadlaq

**Affiliations:** 1King Abdullah Institute for Nanotechnology, King Saud University, Riyadh 11451, Saudi Arabia; mahamed@ksu.edu.sa (M.A.); hhadlaq@ksu.edu.sa (H.A.); 2Department of Physics and Astronomy, College of Sciences, King Saud University, Riyadh 11451, Saudi Arabia

**Keywords:** nanocomposite, oxidative stress, GSH, autophagic vesicles, non-apoptosis, apoptosis

## Abstract

In this study, a nanocomposite of cerium oxide-zinc (CeO_2_-Zn; 26 ± 11 nm) based on the antioxidant rare-earth cerium oxide (CeO_2_) nanoparticles (NPs) with the modifier zinc (Zn) was synthesized by sintering method and characterized. Its bio-response was examined in human umbilical-vein-derived endothelial (HUVE) cells to get insight into the components of vascular system. While NPs of CeO_2_ did not significantly alter cell viability up to a concentration of 200 µg/mL for a 24 h exposure, 154 ± 6 µg/mL of nanocomposite CeO_2_-Zn induced 50% cytotoxicity. Mechanism of cytotoxicity occurring due to nanocomposite by its Zn content was compared by choosing NPs of ZnO, possibly the closest nanoparticulate form of Zn. ZnO NPs lead to the induction of higher reactive oxygen species (ROS) (DCF-fluorescence), steeper depletion in antioxidant glutathione (GSH) and a greater loss of mitochondrial membrane potential (MMP) as compared to that induced by CeO_2_-Zn nanocomposite. Nanocomposite of CeO_2_-Zn, on the other hand, lead to significant higher induction of superoxide radical (O_2_^•−^, DHE fluorescence), nitric oxide (NO, determined by DAR-2 imaging and Griess reagent) and autophagic vesicles (determined by Lysotracker and monodansylcadeverine probes) as compared to that caused by ZnO NP treatment. Moreover, analysis after triple staining (by annexin V-FITC, PI, and Hoechst) conducted at their respective IC50s revealed an apoptosis mode of cell death due to ZnO NPs, whereas CeO_2_-Zn nanocomposite induced a mechanism of cell death that was significantly different from apoptosis. Our findings on advanced biomarkers such as autophagy and mode of cell death suggested the CeO_2_-Zn nanocomposite might behave as independent nanostructure from its constituent ones. Since nanocomposites can behave independently of their constituent NPs/elements, by creating nanocomposites, NP versatility can be increased manifold by just manipulating existing NPs. Moreover, data in this study can furnish early mechanistic insight about the potential damage that could occur in the integrity of vascular systems.

## 1. Introduction

Composite nanomaterials (NMs)/nanoparticles (NPs) are at the cross-roads of drug targeting, imaging and other aspects in nanobiotechnology [1]. Nanobiotechnology, therefore, has promising potentials in the field of regenerative medicine and cancer treatment [2]. NPs based on metal and rare-earth metal have been attributed to extensive application in the field of catalysis, photoluminescence, degradation of environmental pollution and diagnosis of diseases [3]. Moreover, with the help of existing NPs, nanocomposites consisting of more than two independent NPs or materials in a defined ratio are being developed as novel and improved theragnostic agent. In essence, nanocomposites could be explored in the field of nanomedicine in multiple ways. Nanocomposites based on rare-earth cerium oxide (CeO_2_) NPs have attracted attention due to its desired biological and physical property. NPs of CeO_2_ hold great potential as a novel antioxidant [4]. Another attribute of these NPs that enhances their potential usability is their tunability by other NPs or materials [5]. There are several recent reports published on bio-responses of nanocomposites based on CeO_2_ NPs with modifying component such as graphene oxide (GO) [6,7]. NPs of zinc oxide (ZnO) have also been implicated in potential anticancer therapy [8,9] as are the NMs of GO [6] and reduced GO (rGO) [7]. Therefore, we became curious about the possibility of synthesizing a nanocomposite of CeO_2_ and an element representing ZnO NP (i.e., Zn) and its possible biological response as stated below.

As a matter of ongoing efforts in testing tunability and designing nanocomposite, here, we have, therefore, crafted a nanocomposite that contains NPs of CeO_2_ as the base material and Zn as modifier component due to reasons stated above. Synthesized CeO_2_-Zn nanocomposite was explored for its bio-response in a relevant cell line while also evaluating bio-response of nanocomposite constituent elements for comparison purposes to observe the underlying mechanisms in proper context. To the best of our knowledge, the present study may be the first one that examines the biological property of a CeO_2_-Zn nanocomposite. Recall, NPs/NMs/nanostructures (particulate structures in sizes less than 100 nm at least in one dimension) with potential applications in biomedical science need special attention about their impact on the cross-roads of cytotoxicity, oxidative stress and immune system [10]. Human umbilical-vein-derived endothelial (HUVE) cells are the principal cells that line the vasculature beds and form a barrier against infiltrating bacteria, viruses and other foreign particles [11]. Therefore, HUVE cells are a chosen model for carrying out research to get insight into the components of vascular and immune system [12,13]. Being used as a potential antioxidant or as potential therapeutic agents, these NPs and nanocomposite could interact with several components of vascular and immune system eliciting an undesired response [10]. Along with nanocomposite CeO_2_-Zn, NP of CeO_2_ was chosen in this study since it is the base material in the nanocomposite whereas NP of ZnO was chosen as representative form of the modifier component Zn. Therefore, ZnO NP at its IC50 was included in each advanced experiment to compare with ZnO-induced bio-response and understand the factor of Zn in the bio-response induced by the CeO_2_-Zn nanocomposites. Therefore, data in this study can provide important framework about NM/NP tunability and in achieving a more targeted bio-response. Moreover, data in this study can extend mechanistic insight about the potential damage that could occur in the integrity of the vascular system in relevant organisms.

## 2. Materials and Methods

### 2.1. Chemicals and Reagents

Fetal bovine serum, penicillin-streptomycin, calcein-AM, BODIPY and LTR (LysoTracker™ Red DND-99) were purchased from Invitrogen Co. (Carlsbad, CA, USA). Gd(NO_3_)_3_∙6 H_2_O and glycine, DMEM F-12, MTT [3-(4,5-dimethyl thiazol-2-yl)-2,5-diphenyl tetrazolium bromide], NADH, pyruvic acid, perchloric acid, DCFH-DA, Dihydroethidium (DHE), DAR-2, MDC (monodansylcadaverine), autophagy kit, JC-1 (5,5′,6,6′-Tetrachloro-1,1′,3,3′-tetraethyl-imidacarbocyanine iodide), Hoechst (bisBenzimide H 33342 trihydrochloride), PI (3,8-Diamino-5-[3-(diethylmethylammonio)propyl]-6-phenylphenanthridinium diiodide), GSH, o-phthalaldehyde (OPT), hank’s balanced salt solution (HBSS) and Bradford reagent were obtained from Sigma-Aldrich (Sigma-Aldrich, Saint Louis, MO, USA). AnnexinV-FITC apoptosis/necrosis kit was purchased from BD Biosciences USA (Franklin Lakes, NJ, USA). Ultrapure water was prepared from a Milli-Q system (Millipore, Bedford, MA, USA). All other chemicals used were of reagent grade.

### 2.2. Preparation and Characterization of Cerium Oxide-Zn (CeO_2_-Zn) Nanocomposite

Nanocomposite of CeO_2_-Zn was synthesized by mechanical milling followed by sintering process based on the methodology described by Khan et al. [5]. Briefly, nanocomposites with 5% Zn content were prepared by commercially available high purity grade Zn and CeO_2_ nanopowders purchased from Sigma-Aldrich, MO, USA in order to avoid any kind of impurities and, thus, reproducibility-related errors. These nanopowders were put in a high-energy planetary ball mill (PM 100, Retsch, Germany) at room temperature with a ball to powder ratio of 10:1 at the milling speed of 400 rpm for 6 h. After this, the nanocomposite that formed was characterized by field emission transmission electron microscopy (FETEM, JEM-2100F, JEOL, Inc., Tokyo, Japan). Powder x-ray diffraction (PXRD) technique was used for crystallinity characterization using Cu-Kα radiations of wavelength 1.54060 Å. XRD data were collected in the 2θ range of 20 to 80ο at a scan rate 0.02°/s. Hydrodynamic properties of the nanocomposite were determined by the use of a dynamic light scattering system (DLS) (Malvern Instruments, Malvern, UK) according to the method described by Murdock et al. [14]. Nanocomposite was freshly suspended in relevant aqueous media before its DLS measurements at a concentration of 150 µg/mL which is a concentration around which most cellular experiments were carried out.

### 2.3. Cell Culture and Treatment with Various NPs

Human umbilical vein endothelial cells (HUVECs) (ATCC, Manassas, VA, USA) were cultured in DMEM-F12 medium. This medium was completed with the addition of 10% FBS, antibiotic/antimycotic solution (100 U/mL penicillin and 100 µg/mL streptomycin) and endothelial growth components (CADMEC, Cell Applications, Inc., San Diego, CA, USA). Cells were placed in a humidified tissue incubator (HERACell 150i, ThermoFisher Scientific, Waltham, MA, USA) set at 37 °C of temperature and 5% of CO_2_ gas. The cells were passaged before they reach 70–80% of confluency occurring at 3–4 days of interval. Dry forms of NPs were poured directly in the culture media and final concentrations were adjusted and ultrasonicated for 5 min (UltrasonicCleaner-8891, Cole-Parmer, 625 Bunker Court, Vernon Hills, IL, USA) before exposure to cells that were seeded before 24 h. All of the previous media were carefully removed before exposures. All of the biological data provided here represent 24 h of exposure. Nanocomposites and ZnO NPs were selected at their IC50s whereas biocompatible CeO_2_ NP at 100 µg/mL for advanced studies. Untreated HUVE cells served as the control in each experiment.

### 2.4. Cell Viability Assays

Cell viability due to 24 h exposure of various NPs preparations was evaluated by a biochemical protocol reported by Mosmann [15]. Approximately 1 × 10^4^ HUVE cells were treated with NPs for a 24 h duration and an MTT test was run. Briefly, MTT was dissolved in HBSS without phenol red at a concentration of 0.5 mg/mL and filtered with 0.22-micron sterile syringe filter (Sigma-Aldrich (Sigma-Aldrich, Saint Louis, MO, USA). Control and treatment media (with NPs) were removed next day exactly after 24 h and a 200 µL MTT solution prepared in HBSS was added and left for 1.5 h allowing the reduction of MTT to formazan crystal by viable cells. Formazan crystal that formed was solubilized in 20% SDS in 50% dimethyl formamide. Optical density was recorded at 570 nm in a plate reader (Synergy HT, Bio-Tek, Winooski, VT, USA) in a clear NPs-free formazan supernatant that was prepared by centrifugation of 96-well plate at 2147× *g* in a refrigerated centrifuge. Data on cytotoxicity are provided as percentage of control cells where cell viability was considered to be 100% in untreated control cells.

### 2.5. Evaluation of Cell Membrane Integrity

Intactness in the membrane of cells was analyzed by biochemical methods followed by a microscopic imaging on cell morphology. Therefore, membrane integrity was determined by quantifying the amount of lactate dehydrogenase (LDH) release outside in the surrounding media. It was determined in a total volume of 3.0 mL reaction volume carried out in quartz cuvette. Reaction was started by 100 µL of culture media in a mixture that contained 100 μL of 6 mM Na-pyruvate, 100 μL of 0.4 mM NADH and 2.7 mL 0.1 M potassium-phosphate buffer, pH 7.4) [16]. Absorbance of NADH at wavelength of 340 nm at room temperature was recorded for 3 min in spectrophotometer (Genesys 10 Bio, Thermo Fisher Scientific, Madison, WI, USA). LDH activity was provided as the percentage of control. The lipid peroxidation (LPO) reaction with various ROS occurring in fatty acids in the lipid bilayer of cell membranes was quantified by determining the level of adduct formed between thiobarbituric acid (TBA) probe and malonaldehyde (MDA). The resultant TBA-MDA adduct was expressed mainly as thiobarbituric acid reactive substance (TBARS). TBARS value is provided as TBARS/mg protein that was quantified according to the protocol described by Ohkawa et al. [17] using molar extinction coefficient of 1.56 × 10^5^ M^−1^ cm^−1^ for the MDA-TBA adduct. LPO was also determined by live cell imaging under lipophilic C11-BODIPY581/591 (Invitrogen, Carlsbad, CA, USA) probe as described elsewhere [18,19]. This probe emits green fluorescence in the direct proportion of LPO occurred in lipophilic sites of cell membranes while emitting uniform red fluorescence which is irrespective of LPO [20]. Briefly, cells were labelled with the dye at the final concentration of 2 µM in a 12-well culture plate for 60 min. After 60 min, excess dye was carefully removed by washing with HBSS before imaging under a fluorescence microscope (Leica DMi8 manual, Wetzlar, Germany).

### 2.6. Assay of Intracellular H_2_O_2_ and O_2_^•−^

The superoxide radical (O_2_^•−^) and H_2_O_2_ are the species of ROS that have been described most for activating signaling pertaining to survival and death fates [21,22]. Cellular H_2_O_2_ was determined by cell-permeable DCFH-DA (Sigma-Aldrich, Saint Louis, MO, USA) fluorescent probe that upon reacting with H_2_O_2_ produces cell-impermeable fluorescent product (DCF) [23]. Briefly, experimental cells in 96-well plate were stained with 100 µL of 50 µM dye for 45 min. Resultant DCF fluorescence intensity was measured in a plate reader and data provided in the reference of control (Synergy HT, Bio-Tek, Winooski, VT, USA). DHE (dihydroethidium) is a cell-permeable dye that specifically reacts with O_2_^•−^ generating red fluorescent products of ethidium and/or 2-hydroxyethidium [24,25]. These products intensely stain the nucleus due to their strong binding with DNA molecules. Cells were stained with 5 µM DHE for 30 min. In our experience, this dye does not exhibit the phenomenon of auto-fluorescence and, therefore, does not require much washing before imaging. However, we washed it once with HBSS before imaging under a fluorescence microscope using suitable filter cube (Leica DMi8, Wetzlar, Germany).

### 2.7. Detection of Intracellular Nitric Oxide (NO)

NO was evaluated by live cell imaging using a NO-specific cell-permeable dye known as DAR-2. This is a rhodamine-based dye that produces intense fluorescence in infra-red region and cells with high NO appears yellowish in images [26,27,28]. DAR-2 was applied at 10 µM of final concentration prepared in HBSS buffer and incubated for 2 h. Imaging was performed After washing cells 3 times with HBSS, imaging was conducted under a microscope using its red emission filter cube (Leica DMi8, Wetzlar, Germany). NO determination using DAR-2 live cell imaging has been reported by several investigators in cells as well as in vivo models [29,30,31]. A conventional method of measuring NO was also used that quantifies nitrite in cell culture by Griess reagent at wavelength of 540 nm in a plate reader (Synergy HT, Bio-Tek, Winooski, VT, USA) [32,33]. Data have been presented as the percentage of NO assuming NO 100% in control cells.

### 2.8. Measurement of Intracellular GSH

Antioxidant GSH was determined in cell lysate by reacting it with o-phthalaldehyde (OPT, 1 mg/mL in methanol) that produces a fluorescent adduct [34]. Cell lysate was prepared by placing cell pellet in 100 µL lysing solution (0.1% deoxycholic acid and 0.1% sucrose) for 2 h with 3 cycles of freeze–thaw. Lysate was centrifuged at 10,000× *g* for 10 min at 4 °C. Supernatant was precipitated for proteins in 1% perchloric acid to avoid the inclusion of thiols present in proteins and then GSH was determined by OPT reaction. Protein was measured in unprecipitated supernatant and GSH was presented as GSH nmol/mg cellular protein.

### 2.9. Assay of Mitochondrial Membrane Potential (MMP)

Rhodamine-based probe Rh123 (Sigma-Aldrich (Sigma-Aldrich, Saint Louis, MO, USA) was utilized in determining the MMP in cells. Under live cell imaging, this probe gives bright green emission in healthy cells that is faded in the loss of MMP [35,36]. Rh123 was applied at 5 µM concentration prepared in HBSS after removing previous media and treatment agents. After 20 min of incubation with the dye, cells were washed two times with dye-free HBSS and imaging was performed under a fluorescence microscope (Leica DMi8, Wetzlar, Germany).

### 2.10. Analysis of Autophagy

Commencement of autophagic vesicles that typically takes place during autophagy was determined by two dyes—lysotracker red (LTR) and monodansylecadeverine (MDC)— and co-labelled simultaneously. LTR detects acidic vesicles in cells such as intact lysosomes as well as fusion product of lysosomes with autophagosomes (i.e autolysosomes) [37]. MDC detects preferentially autolysosomes without labelling lysosomes qualifying MDC a specific probe detecting the late stage in autophagy [38]. LTR and MDC were multiplexed in HUVE cells at 1 µM and 50 mM, respectively, for 40 min all prepared in HBSS. Images were captured in violet filter cube for (blue) MDC and green filter cube for (red) LTR under the fluorescence microscope.

### 2.11. Determination of Mode of Cell Death by AnnexinV-FITC, PI and Hoechst Staining

Mode of cell death was determined by multiplexing the same cells with three different probes annexinV-FITC, PI and Hoechst 33442. AnnexinV-FITC preferentially binds with the outer leaflet of cell membranes characteristic in early apoptosis. PI preferentially accumulates in cells dying due to non-apoptosis mode of cell death although PI enters even in apoptotic cells but in late stages marking “secondary apoptosis” but not true apoptosis [39]. Hoechst labels dimmer to healthy cell nuclei whereas brighter and punctate in apoptotic nuclei and also brighter necrotic nuclei, but chromatin appears diffused [40]. Recall, nucleus perimeter is dilated and chromatin diffused in necrosis whereas it is shorter and, consequently, more condensed in apoptosis [40,41]. It was carried out according to the protocol described by the kit supplier (B.D. Biosciences, San Jose, CA, USA).

### 2.12. Caspase Assay

Activities of caspases 3 and 9 were determined from cell lysate of control and treated cells. In brief, 5 × 10^4^ HUVECs were seeded in T25 culture flasks and treated with NPs for 24 h. After the completion of treatment for specified duration, cells were washed two times with HBSS buffer and finally collected in 1.5 mL tubes by centrifugation at 4 °C. A reaction mixture containing 30 μL of cell lysate, 20 μL either from Ac-DEVDAFC (caspase-3 substrate) or from Ac-LEHD-AFC (caspase 9 substrate) and 150 μL of protease reaction buffer (50 mM Hepes, 1mM EDTA, and 1mM DTT, pH 7.2) was incubated. Reaction progress was monitored by recording fluorescence at suitable band pass filter at every interval of 5 min for a total duration of 20 min in a plate reader (Synergy HT, Bio-Tek, Winooski, VT, USA). Activities of caspases are given in %age of control.

### 2.13. Protein Estimation

Total protein in cellular lysate was determined by a protein assay kit purchased from the Sigma-Aldrich, Saint Louis, MO, USA.

### 2.14. Statistics

ANOVA (one-way analysis of variance) was applied followed by Dunnett’s multiple comparison tests for statistical calculations. A burst of cellular images was taken by Leica DFC450 camera (Leica Microsystems GmbH, Wetzlar, Germany) at constant exposure time, gain, color saturation and gamma adjustment. CTCF (corrected total cellular fluorescence) was calculated in the ImageJ software (NIH, Bethesda, MD, USA). Mean cellular fluorescence got subtracted by mean background (non-cellular) fluorescence. Specific statistical statements are given in respective figure legends.

## 3. Results

### 3.1. Physicochemical Properties of CeO_2_-Zn Nanocomposite

Average size of the nanocomposites prepared in this study was found to be in the range of 26 ± 11 nm when characterized by the collective analyses of TEM (Figure 1A), high-resolution TEM (Figure 1B), and XRD (Figure 1C). It is worth noting that the precursor NPs (CeO_2_ and Zn) were obtained from commercial sources (Sigma-Aldrich, MO, USA) to avoid any possibility of contamination and impurity in the synthesized nanocomposite. TEM mainly exhibits cubes and partially spherical shapes. The results on size and crystallinity are well aligned with the results obtained from the XRD data (average crystallite size; 23 ± 9 nm). DLS measurement was carried out to see the effects of serum proteins and other media components on the dispersion of the nanocomposite in complete culture medium. Zeta potential of nanocomposite freshly suspended in complete culture media (−37 ± 4 mV) was found to be much fair than it was in pure water (−21 ± 3 mV). This difference resulted in less agglomeration of nanocomposite in complete culture media (147 ± 56 nm) than in pure water (204 ± 73 nm). Aerodynamic and hydrodynamic properties have been summarized in Table 1.

### 3.2. Cytotoxic Potential of CeO_2_-Zn Nanocomposite in HUVE Cells

Cytotoxicity potential due to nanocomposites occurred in a concentration-dependent manner (Figure 2A) that came out to be lesser than it was due to ZnO NPs. As expected, NPs of CeO_2_ did not significantly affect cell viability up to the 200 µg/mL when exposed for 24 h (data not shown). Here, we have chosen 100 µg/mL for CeO_2_ NPs and IC50 concentration for ZnO NPs in studying the advanced biomarkers due to IC50 of CeO_2_-Zn nanocomposite. Nanocomposite of CeO_2_-Zn caused 50% cell growth inhibition at a concentration of 154 ± 6 µg/mL (IC50) when exposed for 24 h while IC50 of ZnO NP for a 24 h exposure in HUVE cells was found to be 78 ± 5 µg/mL. As expected, live cells’ calcein-AM fluorescence is significantly diminished in cells under toxic treatments due to nanocomposites and ZnO NPs (Figure 2B,C). Morphology of cells treated with CeO_2_, as shown in respective phase-contrast microscopic images in Figure 2B also reveals a positive effect of CeO_2_ NPs on the health of cells as depicted by their more flattened nature (i.e., a firmer attachment with culture vessel bottom) of growth in comparison of control cells. Moreover, in ZnO-treated images, membrane blebs and cell shrinkage can be seen that are classical morphological hallmarks of apoptosis. These membrane blebs are also visible under corresponding calcein-AM fluorescence images. Such apoptotic features are almost lacking in cells treated with CeO_2_-Zn nanocomposite underlying a death mechanism that appears independent of apoptosis.

### 3.3. Nanocomposites Had Less Oxidative Potential Than NPs of ZnO

Nanocomposites had significantly lesser impact on membrane damage as compared to that of ZnO but led to high induction of NO and autophagic vesicles in comparison with ZnO NPs as discussed below. Oxidative stress due to CeO_2_-Zn nanocomposite as measured by BODIPY imaging (Figure 3A,B) and TBARS quantification (Figure 3C) and the resulting membrane damage assessed by LDH release (Figure 3D) was lesser for ZnO NPs at their respective IC50s. As expected, NPs of CeO_2_ did not have appreciable effects on LPO and membrane damage. LPO was quantified to be 187% due to CeO_2_-Zn nanocomposite and 293% due to ZnO NPs when determined by BODIPY imaging and 130% and 216%, respectively, when determined by TBARS method. Plasma membrane damage caused by LPO therein is directly proportional to the intensity of BODIPY green fluorescence [19,20]. Similarly, LDH was induced to 134% and 189%, respectively, by nanocomposites and ZnO NP treatments.

### 3.4. Nanocomposite and ZnO NPs Had Complex Effects on ROS Induction and MMP Loss

NPs of ZnO significantly induced ROS when measured by DCF fluorescence that gives a rough quantification of H_2_O_2_ mostly. This ROS (i.e., H_2_O_2_) was much higher for ZnO NPs than the nanocomposite while least for CeO_2_ NPs (Figure 4A). Cells were also measured for another ROS, O_2_^•−^, by a specific probe DHE since this ROS is also implicated in wide variety of cellular events. Due to wide gap in emission spectra, DHE-labelled cells were also co-labelled with Rh123 to measure O_2_^•−^ and MMP simultaneously (see respective cell images of DHE in red color and Rh123 in green in Figure 4B). Cells were merged as a proof of concept as it is carried out in case of a probe with two different emission spectra and/or in case co-imaging with two or more probes with non-overlapping spectra. Merging also provides a logical background of relative comparison and calculations while having many variables constant under the same experimental settings. Fluorescence for DHE and Rh123 is plotted in Figure 4C,D, respectively. Nanocomposite significantly induced O_2_^•−^ radical as evidenced by DHE-labelled cellular image (Figure 4B) DHE fluorescence (Figure 4C) while ZnO seems to primarily induce H_2_O_2_ as suggested by DCF fluorescence (see Figure 4A). It should be noted that CeO_2_ NPs also induced O_2_^•−^ at non-toxic concentrations as well as slightly less DCF fluorescence. O_2_^•−^ induced at the highest concentration was due to the nanocomposite. Interestingly, ZnO causing high induction in DCF intensity did not induce DHE as significantly from that of the control.

### 3.5. Nanocomposite Had Higher NO Inducing Potential than NPs of ZnO

Nanocomposite caused significant induction of NO in HUVE cells while ZnO NPs did not cause NO to be induced at appreciable extent when measured by NO-specific fluorescent dye DAR-2 (see fluorescent images under ‘DAR-2′ in Figure 5A and its quantification in Figure 5B). NO was also determined in HUVE cells due to similar treatments by indirectly quantifying nitrite by Griess reagent (Figure 5C). Data suggested that toxic ZnO treatment caused the lowest NO production among all treatments whereas CeO_2_ NP treatment causes the highest NO production. In other words, nanocomposite treatment still induced NO higher than that seen in the ZnO treatment.

### 3.6. Nanocomposites Induced Higher Autophagy and GSH Depletion than ZnO NPs

In the present study, autophagy, as a function of autophagic vesicles detected tandemly by LTR and MDC probes [42] (see corresponding images in Figure 6A and their fluorescence quantifications in Figure 6B,C, respectively), was found to be significantly elevated due to the CeO_2_-Zn nanocomposite in comparison with the control as well as the ZnO NP-treated cells. Compared to the nanocomposite, to a lesser extent, ZnO treatment also led to autophagy induction compared to the control cells. Antioxidant GSH depletion was the most significant for the nanocomposite while ZnO also significantly exhausted cellular GSH (Figure 6D). Buthionine-(S,R)-Sulfoximine (BSO) has been used as a positive control for the GSH depletion in cells as it is a potent inhibitor of a rate-limiting γ-glutamylcysteine synthetase enzyme in GSH bio-synthesis [43].

### 3.7. Mode of Cell Death Due to Nanocomposites Was Apoptosis-Independent

The induction of ROS and loss of MMP due to ZnO NPs lead to apoptosis as revealed in triple-staining (annexinV-FITC/PI/Hoechst 33442) analysis under fluorescence microscopy (Figure 7A). Triple-stained imaging studies also confirm a mode of cell death by CeO_2_-Zn to be different from that induced by ZnO NPs. Bigger circles in nanocomposite-treated cells shows heavy staining by PI (red nuclei) while almost lacking annexinV-FITC (green) staining in corresponding cells. Bigger circles in the case of ZnO exposure represent that a staining pattern occurs for both PI as well as annexinV-FITC, a kind of staining pattern generally considered in apoptotic cells. Group of cells enclosed by smaller circles in each treatment group represent an opposite pattern to that of the major pattern represented in the bigger circle and represent deviation against the major events in their respective treatments. Same color of circles represents “same process” and “bigger” circles represent major events while “small” circles represents rare events, respectively, in their own treatment conditions. For example, red circle (representing non-superimposable PI and annexinV-FITC staining) is bigger for the CeO_2_-Zn nanocomposite treatment while it is much smaller in the ZnO treatment. Similarly, yellow circle (representing “superimposability” of PI and annexinV-FITC staining) characteristic of apoptosis is much bigger in the ZnO treatment whereas it is much smaller (and, thus, rare mode of cell death) in the nanocomposite treatment. Caspase 9 and 3 are crucial initiator and effector enzymes responsible for the onset of apoptosis in cells undergoing irreparable damages due to oxidative stress. The activities of these enzymes further help in deciphering the process of apoptosis from that of non-apoptosis. Caspase 9 is known as initiator caspase required in initiating the intrinsic or mitochondrial pathway of apoptosis [44,45]. In the present study, caspase 9 was significantly activated due to NPs of ZnO while no appreciable change in activity was detected due to NPs of CeO_2_ and nanocomposite of CeO_2_-Zn as compared to the control (Figure 7B). As caspase 3 is one of the several effector caspases activated in apoptosis by caspase 9, its activity was significantly higher due to NPs of ZnO while no appreciable change in activity was detected due to NPs of CeO_2_ (7C). Although caspase 3 activity caused by CeO_2_-Zn was significantly high as compared to the control, it was much lower than that induced by ZnO NPs (see Figure 7C). From morphological (see phase contrast images in Figure 2B) to triple staining (see Figure 7A) to enzymatic activities of caspases (see Figure 7B,C), it is clear that ZnO activates an apoptotic mode of cell death whereas the nanocomposite of CeO_2_-Zn activates a non-apoptotic mode of cell death.

## 4. Discussion

CeO_2_ NPs has attracted significant attention in nanomedicine owing to its promising potential in the therapy of diseases arising out of oxidative stress. The most reactive rare earth NPs, the NPs of CeO_2_, are known for possessing high band gap energy (E_g_ ~ 3.19 eV) and extraordinary dielectric properties (ε  =  24.5) [5]. Due to high reactivity, CeO_2_ NPs provide the opportunity for fine tuning with other materials and elements of interest (such as Zn, Fe, Cu, etc.) generating a variety of nanocomposites with the potential to invoke bio-response differently from either of the constituent materials [5]. Average size of nanocomposites of CeO_2_-Zn prepared in this study was found to be in the range of 26 ± 11 nm that matched closely with the crystallite size (23 ± 9 nm) provided by XRD. As data suggested, the degree of protein adsorption to nanocomposite surface in aqueous media resulted in a significantly fair distribution in the complete culture media as compared to the nanocomposites suspended in pure water. NP surface, if highly reactive, can result in different hydrodynamic sizes for the same aerodynamic size of NP when measured in media with different contents of soluble proteins and other components of organic origin [46]. In addition, to modifying hydrodynamic sizes in solution, surface modifications can impart stealth features to the NPs/nanostructures in the course of recognition by the immune system [47].

There appears a discrepancy in the reports about biological activity of nanocomposites based on CeO_2,_ as some studies cited below for example. Nanocomposites synthesized with CeO_2_ and graphene oxide (GO) induced much higher toxicity in A549 cells in comparison with either of the co-materials (i.e., CeO_2_ and GO) used in the synthesis of the CeO_2_-GO nanocomposite [6]. Moreover, CeO_2_-GO nanocomposite elicited a mode of cell death different from that induced by GO NPs [6]. Interestingly, the nanocomposite of CeO_2,_ when modified with rGO (reduced graphene oxide), was found less toxic than the NPs of rGO alone in human lung epithelial (A549) cells [7]. Mechanistically, GSH-replenishing tendency of CeO_2_ in the CeO_2_-rGO nanocomposite was explained for the apparent lesser toxicity due to CeO_2_-rGO nanocomposite in A549 cells. It is also worth noting that NPs of GO are more toxic than NPs of r-GO [7]. These two CeO_2_-based nanocomposites thus impart toxicity in accordance with the toxic potential of modifying material; the higher the toxic potential of the modifying component, the more severe the toxicity is due to the resultant nanocomposite. NPs of Ag (silver) are known for their intense toxicity and excellent antibacterial properties [48,49]. As expected, Ag doping significantly increased cytotoxicity of otherwise biocompatible magnesium oxide (MgO) NPs in HUVE cells according to the amount of the Ag present in the MgO-Ag nanocomposite [50]. Reports suggests that rGO NPs are less toxic than NPs of Ag alone [51]. Contrary to the notion above, nanocomposites of Ag-rGO did not moderate the toxicity observed for either NPs of Ag or r-GO; instead, this nanocomposite induced significantly higher toxicity than the toxicity of either constituent (i.e., rGO andAg) [51]. In fact, cytotoxicity due to the Ag-rGO nanocomposite came out to be 2-fold higher than the most toxic constituent NPs of Ag when exposed in the two human cancer cells originating from the breast (MCF-7 cells) and lung (A549 cells) [51]. Importantly, this nanocomposite (Ag-rGO) turned out to be less toxic in normal breast (MCF-10A cells) and normal lung fibroblasts (IMR-90 cells) suggesting nanocomposite crafting might be used as an alternative in generating a nanostructure with novel anticancer potential from existing NPs that are not exhibiting anticancer potential [51]. Again, the nanocomposite of molybdenum/zinc oxide/rGO (Mo/ZnO/rGO) was found to be 3 times more highly efficient than ZnO NPs in bringing the demise of human colon cancer HCT116 cells and breast MCF7 cancer cells [52]. Moreover, biocompatibility of Mo/ZnO/rGO nanocomposites in human normal colon epithelial NCM460 cells and normal breast epithelial MCF10A cells was much better than those of pure ZnO NPs. These nanocomposites provide opportunity for exploration on grounds of efficient anticancer activity as well as safety potential in normal cells [52]. In this study, the nanocomposite induced less toxicity than toxic NPs of ZnO, a finding which is in accordance with CeO_2_-based nanocomposites as discussed above. However, when advanced parameters of toxicity were evaluated, the mechanism of toxicity and, consequently, the mode of cell death in HUVE cells due to the CeO_2_-Zn nanocomposite turned out to be quite different from that of toxic ZnO NPs.

Cytotoxic NPs are well known for causing loss in MMP which is often considered a committed step in initiating programmed cell death [53]. In this study, ZnO led to less production of O_2_^•−^ in comparison with the CeO_2_ and CeO_2_-Zn nanocomposite. ZnO NPs, however, caused a higher loss in MMP as compared to the nanocomposite (see Figure 4B,D). CeO_2_ NP treatment, however, significantly caused a gain in MMP in comparison with control cells (see Figure 4B,D). The data in this study clearly suggest that CeO_2_ NP treatment has no deleterious effect on MMP in spite of inducing O_2_^•−^ in HUVE cells. Similar to ROS, NO and its bye-products may lead to cell damage in multiple ways [54,55]. It has been suggested that high levels of NO might be damaging to endothelial cells via inducing necrosis followed by energy-depletion-induced autophagy [54,55]. It has been proposed that high concentrations of NO can activate necrosis followed by energy depletion and autophagy or can activate apoptosis followed by persistent oxidative stress and loss of MMP [54,55]. Upon reaction with various members of ROS, particularly with O_2_^•−^, NO can form other kinds of reactive nitrogen species (RNS) with more damaging potential than NO and ROS themselves [56,57]. Findings on NO measurement in this study revealed a strong NO modulation by different nanomaterials examined here. In the current study, CeO_2_ NPs induced high NO without inducing cytotoxicity, whereas the nanocomposite CeO_2_-Zn induced NO as well as significant toxicity. Moreover, the most toxic ZnO NP did not have any significant effect on the NO level in the cells used in this study. In this study, a higher intensity in DAR-2 fluorescence was recorded due to the biocompatible CeO_2_ NP treatment whereas a lower intensity was recorded for the nanocomposite. Alternatively, membrane-damage-inducing CeO_2_-Zn nanocomposite led to a higher capture in nitrite in the surrounding culture media when measured by Griess reagent as compared to that for CeO_2_ NPs. This discrepancy can be partly explained by the fact that NO diffusion could be expected to occur with a lower rate across an intact membrane than it would be across a damaged one. Therefore, in cells that had been induced for NO production, the concentration of NO available to react with DAR-2 could be higher in cells having an intact membrane (as in CeO_2_ NP treated cells) than in cells with a damaged membrane [26,27,31]. Additionally, it should be realized that not all NO produced in cells is necessarily leaked out of the cells to be converted into nitrite and nitrate. NO can find several ways to be consumed within cells before being diffused out of cells [30,31]. NO concentration in cells is subject to change which is dependent on several factors. Factors such as NO stability, its local diffusion through membranes, consumption into reactions leading to the formation of other reactive nitrogen species (RNS) (such as peroxynitrite), other substrate availability (such as its faster-than-diffusion reaction with iron-haem in oxyhaemoglobin if in vivo) as well as the type of cells and redox status of the cell producing it are known to affect NO levels in cells [29,56,57]. The two methods utilized here for determining NO levels in cells should therefore be viewed as complementary and robust in that they clearly provide evidence of NO production due to cerium-containing nanomaterials but not by ZnO NPs in HUVE cells. Consequently, NO induction or inhibition can result in dramatic effects for CeO_2_ NP, CeO_2_-Zn nanocomposite and ZnO NPs. Rare-earth nanomaterials of gadolinium, cerium and neodymium have been reported to induce autophagy in a wide variety of cell types [58]. Data in the present study suggested that the toxic CeO_2_-Zn nanocomposite induced high NO and autophagic vesicles that was much higher than that induced by ZnO NPs, clearly suggesting a different mechanism of toxicity by the two nanomaterials. Antioxidant GSH is a ubiquitous non-enzymatic molecule that protects cells from the damages occurring by harmful action of ROS/RNS [59]. NPs with the tendency to induce oxidative stress led to significant exhaustion of cellular antioxidants [19,20] as was severely depleted by the NP of ZnO and CeO_2_-Zn in this study. NPs of antioxidant CeO_2_, on the other hand, caused a GSH increase in comparison with the control cells, suggesting its biocompatibility and antioxidant property in HUVE cells.

In this report, cells treated with the CeO_2_-Zn nanocomposite are heavily stained with PI but significantly lacking annexin binding, a phenomenon not considered in apoptosis. Early apoptosis is represented by annexinV binding whereas in late apoptosis (or secondary necrosis), PI also accumulates in the nuclei of cells undergoing apoptosis [60,61]. Caspase 9 is activated within the apoptosome that is formed as a result of apoptotic signals such as cytochrome c release in cytosol from damaged and, thus, leaky outer membrane of mitochondria [62]. Once activated, caspase 9 facilitates apoptosis via activating many downstream effector caspases such as caspase 3, 6 and 7 [45]. In addition to bringing apoptosis, caspase 9 also plays a role in many non-apoptotic functions such as cell differentiation, innate immunity, mitochondrial homeostasis and autophagy [45]. In the present study, caspase 9 was significantly activated due to NPs of ZnO while no appreciable change in activity was detected due to NPs of CeO_2_ and nanocomposite of CeO_2_-Zn. Autophagy has been considered to be an important mechanism of cell survival under conditions of stress and continued nutrient starvation by recycling cellular macromolecules and organelles [63]. Cell death, of course, would be followed if pro-survival conditions are not restored and the type of resulting cell death is often called autophagic cell death. This cell death, as a result of autophagic failure, closely resembles a caspase-independent necrosis-like cell death associated with the accumulation of autophagosomes in cells [62]. Non-apoptosis mode of cell death caters a diverse mechanism of cell death that not only includes necrosis but also refers to cell death occurring due to autophagic failure, ferroptosis, programmed form of necrosis known as necroptosis, pyroptosis, mitotic catastrophe, oncosis and so on [64].

## 5. Conclusions

Our findings on advanced biomarkers such as autophagy and mode of cell death suggested CeO_2_-Zn might behave as an independent nanostructure. Since nanocomposites can behave independently of their constituent NPs, by creating nanocomposites, NP versatility can be increased manifold by just manipulating existing NPs. Moreover, added capacity of fine tuning in nanocomposite synthesis greatly increases the likelihood of invoking targeted response in the course of nanomedicine-related research. Moreover, data in this study can extend mechanistic insight about the potential damage that could occur in the integrity of vascular systems.

## Figures and Tables

**Figure 1 toxics-10-00250-f001:**
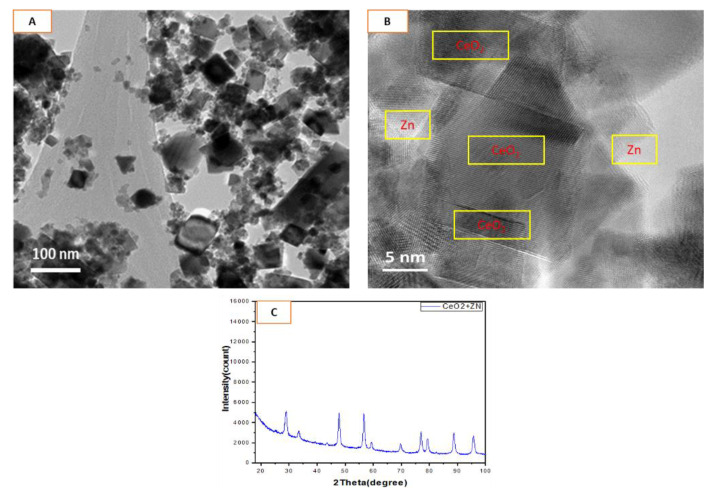
Size, shape and crystallinity characterization of CeO_2_-Zn nanocomposite. (**A**) TEM image captured at 50 nm of resolution and (**B**) captured at 5 nm of resolution (i.e., High-res image). Lighter areas represent Zn whereas relatively darker and larger areas represent CeO_2_ NP in high-res image B. XRD of nanocomposite is given in (**C**).

**Figure 2 toxics-10-00250-f002:**
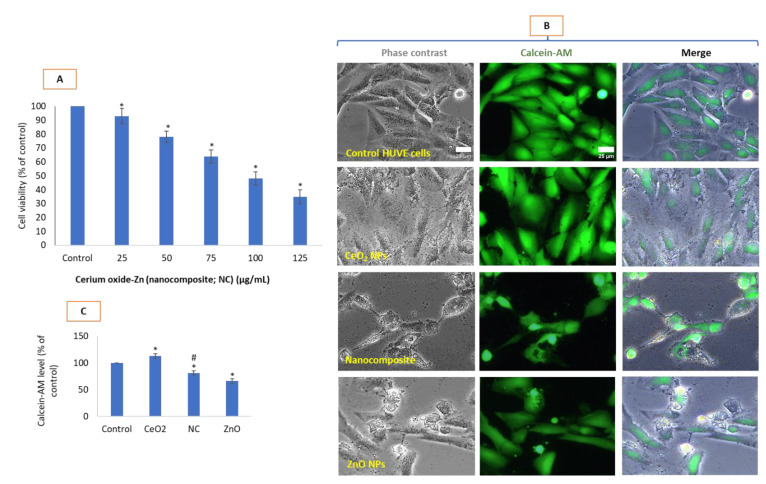
Potential cytotoxicity in HUVECs due to CeO_2_-Zn nanocomposite at concentrations indicated (**A**). Cell morphology under phase contrast in (**B**) and calcein-AM fluorescence (green images in (**B**)) is provided as direct observation of cell damages caused by designated treatments. Calcein-AM fluorescence intensity from calcein-AM images present in Figure 2B is plotted as bar diagram and is given as Figure (**C**). In all subsequent figures of the bar diagram and image, CeO_2_ and CeO_2_ NP both refer to 100 µg/mL of CeO_2_ NPs. NC and nanocomposite both refer to IC50 of CeO_2_-Zn nanocomposite. ZnO and ZnO NP both refer to IC50 of ZnO NPs. In horizontal axis of each bar diagram given, NPs of CeO_2_ and ZnO were denoted simply as CeO_2_ and ZnO, respectively, while the nanocomposite of CeO_2_-Zn as NC due to limited space there. In images, names are in full form. Merge refers to the images obtained after color merging of phase-contrast and calcein-AM images. IC50s calculations were performed using the online IC50 calculator (https://www.aatbio.com/tools/ic50-calculator, accessed on 16 January 2022) provided by AAT Bioquest, Inc. (Sunnyvale, CA, USA). Scale bar represents 25 µm which is only provided in control images as a matter of convention. Each image was captured by a 40× objective under uniform conditions of illumination intensity, time and other variables. Each experiment was performed in triplicates (*n* = 3). * statistically significant difference from the control (*p* < 0.05). # statistically significant difference in between treatments of CeO_2_-Zn nanocomposite and ZnO NP (*p* < 0.05).

**Figure 3 toxics-10-00250-f003:**
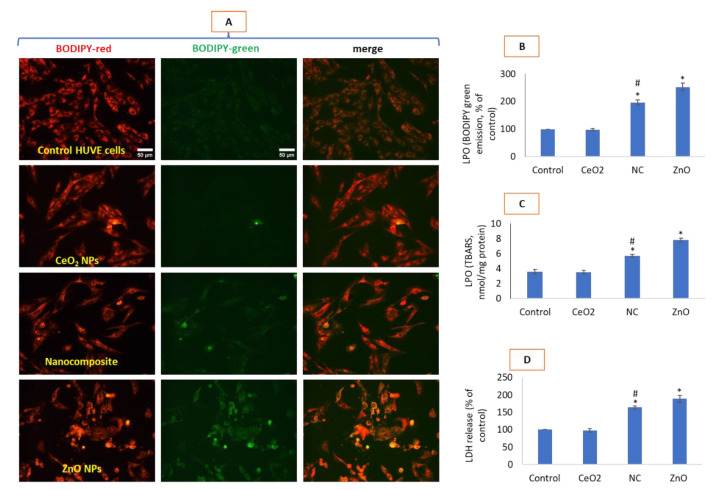
Potential generation of lipid peroxidation (LPO) in the membrane of HUVEC cells due to various NP treatments was evaluated by BODIPY imaging under fluorescence microscopy (**A**), BODIPY green fluorescence intensity quantification (**B**), TBARS quantification biochemically (**C**) and LDH enzyme release in surrounding cell culture media (**D**). Only green fluorescence is plotted as a graph as it is the fluorescence which is modulated in proportion to the amount of LPO occurring in membranes (red fluorescence is uniform in control and treated cells and signals about its quality of staining in cells). Merge refers to the images obtained after color merging of red and green images of BODIPY in ImageJ. Scale bar represents 50 µm and provided only in control images. Each image was captured by a 20× objective under uniform conditions of illumination intensity, time and other variables. Each experiment was performed in triplicates (*n* = 3). * statistically significant difference from the control (*p* < 0.05). # statistically significant difference in between treatments of CeO_2_-Zn nanocomposite and ZnO NP (*p* < 0.05).

**Figure 4 toxics-10-00250-f004:**
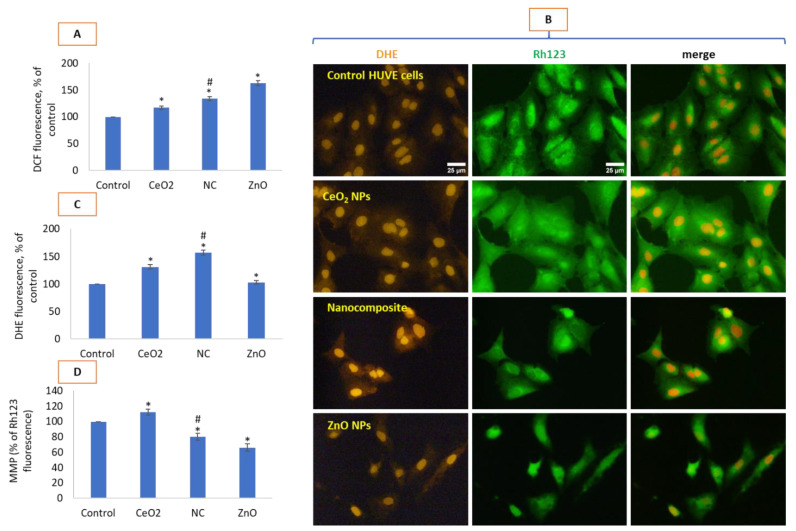
Potential generation of ROS in HUVEC cells due to various NP treatments was evaluated by DCFH-DA (**A**), DHE imaging (red images in (**B**)) and its fluorescence quantification (**C**). MMP was determined by Rh123 imaging (green images in (**B**)) and its fluorescence quantification (**D**). Merge refers to the images obtained after color merging of DHE and Rh123 images in ImageJ. Scale bar represents 25 µm and was provided only in the control images. Each image was captured by a 40× objective under uniform conditions of illumination intensity, time and other variables. Each experiment was performed in triplicates (*n* = 3). * statistically significant difference from the control (*p* < 0.05). # statistically significant difference in between treatments of CeO_2_-Zn nanocomposite and ZnO NP (*p* < 0.05).

**Figure 5 toxics-10-00250-f005:**
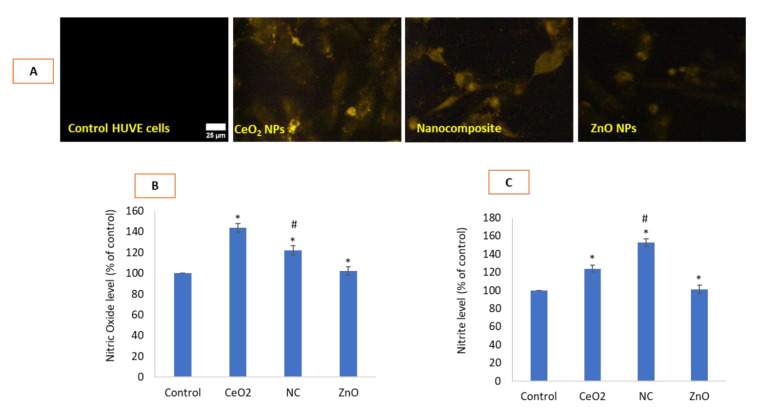
Potential generation of NO in HUVEC cells due to various NP treatments was evaluated by DAR-2 imaging in infra-red region (see images under DAR-2 in Figure (**A**)) and its quantification (**B**). NO was also determined by indirectly quantifying nitrites by Griess reagent (**C**). Scale bar represents 25 µm. Each image was captured by a 40× objective under uniform conditions of illumination intensity, time and other variables. Each experiment was performed in triplicates (*n* = 3). * statistically significant difference from the control (*p* < 0.05). # statistically significant difference in between treatments of CeO_2_-Zn nanocomposite and ZnO NP (*p* < 0.05).

**Figure 6 toxics-10-00250-f006:**
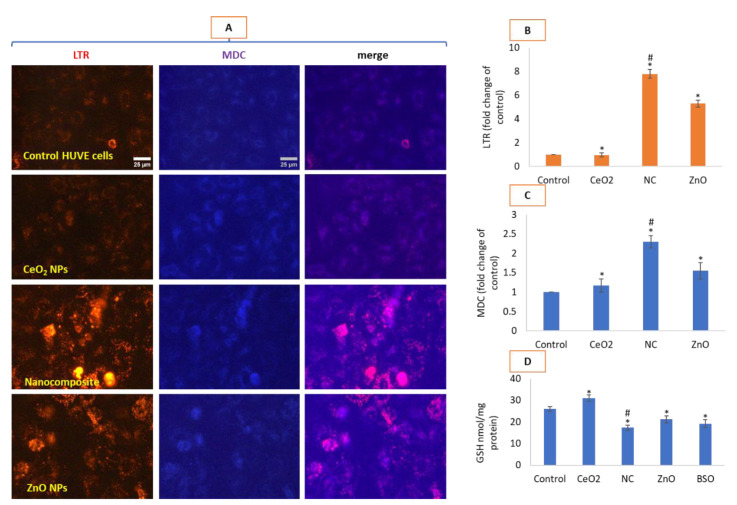
Autophagic vesicles were visualized by LTR (red fluorescence images) and MDC (blue fluorescence images) co-staining in HUVE cells (**A**) treated with NPs for 24 h. Fluorescence quantifications for LTR and MDC, respectively, are given in (**B**,**C**). Merge refers to the images obtained after color merging of LTR and MDC images in ImageJ. Antioxidant GSH modulation due to NPs is given in (**D**) where BSO stands for buthionine-(S,R)-sulfoximine which is an inhibitor of an enzyme involved in GSH synthesis. Scale bar represents 25 µm (40× objective). Each experiment was performed in triplicates (*n* = 3). * statistically significant difference from the control (*p* < 0.05). # statistically significant difference in between treatments of CeO_2_-Zn nanocomposite and ZnO NP (*p* < 0.05).

**Figure 7 toxics-10-00250-f007:**
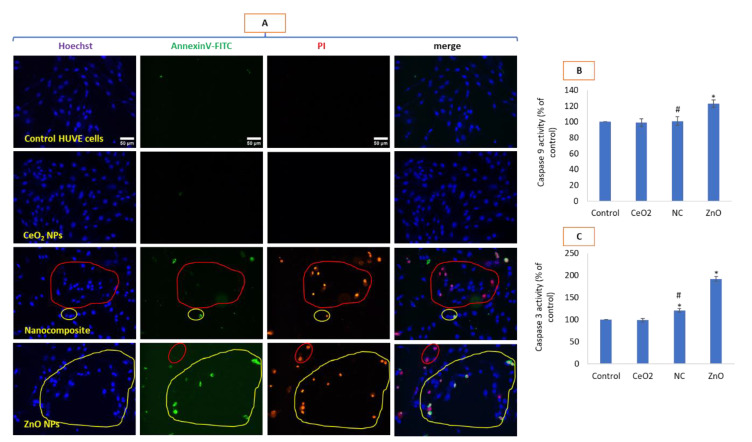
HUVEC cells treated with mentioned NPs for 24 h and potential induction of apoptosis/necrosis was determined by triple staining (**A**), caspase 9 activity (**B**) and caspase 3 activity (**C**). Cells were stained with Hoechst 33442 (blue color) that stains the nucleus of live or dead cells, PI (red color) that stains the nucleus of only dead or dying cells and annexin-V (green color) that preferentially stains apoptotic cells. Dying or dead cells that are stained with PI but lack annexinV-FITC fluorescence clearly undergo a mode of cell death that is not apoptosis. Merge refers to the images obtained after color merging of triple images of Hoechst, annexinV-FITC and PI, respectively, in ImageJ. Scale bar represents 50 µm and is only provided in control images. Each image was captured by a 20× objective under uniform conditions of illumination intensity, time and other variables. Each experiment was performed in triplicates (*n* = 3). * statistically significant difference from the control (*p* < 0.05). # statistically significant difference in between treatments of CeO_2_-Zn nanocomposite and ZnO NP (*p* < 0.05).

**Table 1 toxics-10-00250-t001:** Summary of physicochemical properties of CeO_2_-Zn nanocomposite.

Parameters	Physicochemical Properties
Color	White, powdery
TEM (size)	26 ± 11 nm
TEM (shape)	Mainly cubes and some partially spherical
XRD	Crystalline (average crystallite size; 23 ± 9 nm)
**DLS Values in Complete Culture Media**	
Hydrodynamic size	147 ± 56 nm
Zeta potential	−37 ± 3 mV
**DLS Values in Pure Water**	
Hydrodynamic size	204 ± 73 nm
Zeta potential	−21 ± 4 mV

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
