# Peer review of "CeO_2_-Zn Nanocomposite Induced Superoxide, Autophagy and a Non-Apoptotic Mode of Cell Death in Human Umbilical-Vein-Derived Endothelial (HUVE) Cells"

_toxics, 2022, doi:10.3390/toxics10050250_

Round 1
Reviewer 1 Report
The manuscript is well organized, results are clearly presented and references are up-to-date.
There are only some minor issues that should be addressed by authors.
Line 11: Please add the purpose of the synthesized nanocomposite.
Line 14: If the change is not significant, there is no need to write p-level.
Line 60: Please explain what did you mean by ‘undesired-response’.
Lines 68, 106-107: Please add the references.
Line 134: Please use g, not rpm
Figure 2 caption: Which part belongs to (C)?
Figure 3 caption: Which part belongs to (D)?
Author Response
The manuscript is well organized, results are clearly presented and references are up-to-date.
There are only some minor issues that should be addressed by authors.
Line 11: Please add the purpose of the synthesized nanocomposite.
- Thank you for the encouraging lines. We elaborated the purpose of synthesizing nanocomposite in the revised manuscript. Kindly see rephrased sentences highlighted in red colour. Thanks once again.
Line 14: If the change is not significant, there is no need to write p-level.
- We agree with the reviewer’s point. We have deleted it as per reviewer’s suggestion. Thanks.
Line 60: Please explain what did you mean by ‘undesired-response’.
- By un-desired response we meant an un-intended response; a response deviated from what was thought to occur. We will be happy to remove this word, if the learned reviewer thinks its real meaning is something different or inappropriate. Thanks for this comment.
Lines 68, 106-107: Please add the references.
- For line 68, after a careful observation we restructured that part and we believe now it no longer requires a reference. A reference for DLS method is added. Kindly have look in the revised manuscript. Thanks for the comment.
Line 134: Please use g, not rpm
- Thank you for raising this comment. We apologize for a typical error; it should be 4000 rpm (not as 400 rpm). Now this value (4000 rpm of a plate rotor) is converted to ‘g’ by taking 12 cm (actual is 12.1 cm) radius of swinging plate rotor as specified by the manufacturer. Conversion was done in an online calculator. (https://www.sigmaaldrich.com/SA/en/support/calculators-and-apps/g-force-calculator) Thanks for the comment.
Figure 2 caption: Which part belongs to (C)?
- We thank and appreciate the reviewer’s sharp notice. It is now fixed in light of the reviewer’s comment. Thanks once again.
Figure 3 caption: Which part belongs to (D)?
- We thank and appreciate reviewer’s sharp notice. Actually, we forgot to mention figure 3B (green BODIPY fluorescence intensity diagram) in the concerned legend and figures ended to C. Therefore, it was (and is) LDH that belongs to D. Thanks once again.
We take this opportunity of revision to extend our sincere thanks for the meticulous effort put by the learned reviewers resulting in great improvement of the manuscript under consideration.
Reviewer 2 Report
The study entitled “CeO2-Zn nanocomposite induced superoxide, autophagy and a non-apoptotic mode of cell death in human umbilical vein derived endothelial (HUVE) cells” by Mohd Javed Akhtaraniali et al. synthesized a nanocomposite containing CeO2 and Zn, and examined the effects of this nanocomposite and related nanoparticles on HUVE cells by employing various cytotoxicity tests. Overall, this study was well designed, carefully executed with a set of appropriate experimental procedures, and results properly interpreted. This paper should be of interest to the journal’s targeted audience. However, the reviewer has some major concerns and a couple of minor concerns as outlined below.
Major concern:
(1) The information in Section 2.13 is lacking, compared to other sections in Section 2. There is no procedure included or assay kit information provided.
(2) In Figure 5, the CeO2 NP causes the highest NO level (5A,B) but when measured with a different agent, it is the NC that causes the highest nitrite level (5C). The authors should explain such apparent discrepancy.
(3) In Figure 6D, there is a bar labeled as “BSO”. The reviewer could not find anywhere in the manuscript that explains what "BSO" is and why it is included in this figure.
(4) Figure 7C, there is symbol indicating a significant difference between control and NC, but in the text, page 13 lines 413-416, the interpretation is opposite to what the data show.
(5) In the Discussion, page 14-15, the authors discuss the implication of NO in cytotoxicity and the varying degrees of NO induction by different NPs and NC. However, since the CeO2 NP is showing the least toxicity but also induces NO to a similar level as NC does (Figure 5B,C), is the NO discussion really relevant here?
(6) Overall, there are many abbreviations used in the text which are not spelled out even for the first appearance.
Minor concerns:
There are places that appear to be confusing to the reviewer, which may likely be so to the readers, too. Some examples are listed below.
(1) In the abstract as well as in the main text, terms such as “ZnO”, “ZnO NP”, “nanocomposite”, “CeO2-Zn nanocomposite” are used inconsistently. These terms need to be used in a more consistent way, so the reader knows which specific test material the authors are referring to.
(2) Page 9, the Figure 3 legend does not match the actual subpanels (B-D). As in the legend, there is no (D).
Author Response
The study entitled “CeO2-Zn nanocomposite induced superoxide, autophagy and a non-apoptotic mode of cell death in human umbilical vein derived endothelial (HUVE) cells” by Mohd Javed Akhtar et al. synthesized a nanocomposite containing CeO2 and Zn, and examined the effects of this nanocomposite and related nanoparticles on HUVE cells by employing various cytotoxicity tests. Overall, this study was well designed, carefully executed with a set of appropriate experimental procedures, and results properly interpreted. This paper should be of interest to the journal’s targeted audience. However, the reviewer has some major concerns and a couple of minor concerns as outlined below.
Major concern:
- The information in Section 2.13 is lacking, compared to other sections in Section 2. There is no procedure included or assay kit information provided.
- Section 2.13 (under methodology) mentions about protein estimation to express some value in specific activity. This method is some of the most frequently used methods in science and, therefore, it is written in very short as such in most papers. Learned reviewer is requested to consult few published papers that mention protein estimation about to get on what we mean.
- In Figure 5, the CeO2 NP causes the highest NO level (5A,B) but when measured with a different agent, it is the NC that causes the highest nitrite level (5C). The authors should explain such apparent discrepancy.
- The learned reviewer has raised a very important point. We have tried a possible answer that we can put here but we don’t mean that this is exclusive. Other better explanation, unknown to us, may also be possible.
It is important to realize that not all NO produced in cells are converted in to nitrite and nitrate. Concentration of NO is dependent on many factors such as its local diffusion through membranes, consumption into other reactive nitrogen species (RNS) such as peroxynitrite, substrate availability as well as type of cells and redox status producing it. Moreover, compromised cell membrane could result leakage of more NO in surrounding media than it has been available to react with DAR-2 dye in case of nanocomposite treatment. Therefore, a possible explanation could be that more NO could have been leaked out from cells treated with nanocomposite in comparison of from those cells having more intact membrane as it is in the case of CeO2 NP treatment. In addition, we can see only that fraction of NO which is present inside cells when using DAR-2 dye but we see only that NO measured indirectly as nitrite in culture media when using Griess reagent. The two methods have different principle as well as take different approaches in terms of direct vs indirect measurement and detection sensitivity. Therefore, the methods utilized here should be viewed as complementary in that they clearly provide evidence of NO production in HUVE cells due to CeO2 NP as well as CeO2-Zn nanocomposite. We hope we have made our point clear. Thanks to raising this point as it looks really an important thing from a reader perspective.
- In Figure 6D, there is a bar labeled as “BSO”. The reviewer could not find anywhere in the manuscript that explains what "BSO" is and why it is included in this figure.
We sincerely apologize for not explaining BSO. We thank and appreciate the reviewer’s sharp notice. BSO stands for buthionine-(S,R)-sulfoximine which is often used as positive inhibitor of GSH synthesis that lead to GSH depletion in cells. Now we have mentioned this in the revised version with a proper reference. Kindly have a look. Thanks once again.
- Figure 7C, there is symbol indicating a significant difference between control and NC, but in the text, page 13 lines 413-416, the interpretation is opposite to what the data show.
Thanks goes to reviewer’s sharp notice to this anomaly left during the writing of the manuscript. We have clarified the results on caspase 3 due to CeO2-Zn (it went escaped from our notice for similar sounding CeO2). Kindly, see it in the revision. Thanks once again.
- In the Discussion, page 14-15, the authors discuss the implication of NO in cytotoxicity and the varying degrees of NO induction by different NPs and NC. However, since the CeO2 NP is showing the least toxicity but also induces NO to a similar level as NC does (Figure 5B,C), is the NO discussion really relevant here?
By discussing NO in cytotoxicity (that too in very brief), we meant to emphasize the fact that induction of NO cannot always result in cytotoxicity; rather it may lead to quite opposite end-point as well. Effect of NO must only be viewed in a picture that somehow cater the broad range of modulating activities of NO in a biological environment. We agree with the point of reviewer also here but we think some other reader may feel lacking in the completeness of that ‘picture’ of the very important NO biology. As in the current study for example, CeO2 is inducing high NO but without inducing cytotoxicity whereas NC is inducing NO but with significant toxicity. Moreover, the most toxic ZnO NP goes without having any significant effect on NO level in the cell type used in this study. We thank and respect reviewer’s point of view and hope reviewer be satisfied with our humble opinion.
(6) Overall, there are many abbreviations used in the text which are not spelled out even for the first appearance.
We completely agree with the learned reviewer and we have carefully revised this manuscript from the abstract to conclusion according to reviewer’s suggestion. We would like bring attention of the learned reviewer that throughout in the X-axis of the diagrams we have omitted ‘NP’ and/or ‘nanocomposite’ words due to constrained space there. We have also mentioned this fact in cytotoxicity data figure legend. Thanks once again.
Minor concerns:
There are places that appear to be confusing to the reviewer, which may likely be so to the readers, too. Some examples are listed below.
- In the abstract as well as in the main text, terms such as “ZnO”, “ZnO NP”, “nanocomposite”, “CeO2-Zn nanocomposite” are used inconsistently. These terms need to be used in a more consistent way, so the reader knows which specific test material the authors are referring to.
We completely agree with the learned reviewer and we have carefully revised this manuscript from the abstract to conclusion according to reviewer’s suggestion. We would like bring attention of the learned reviewer that throughout in the X-axis of the diagrams we have omitted ‘NP’ and/or ‘nanocomposite’ words due to constrained space there and we have mentioned this convention appearing at the first place in the legend of figure number 2. Thanks once again.
(2) Page 9, the Figure 3 legend does not match the actual subpanels (B-D). As in the legend, there is no (D).
We thank and appreciate reviewer’s sharp notice. Actually, we forgot to mention figure 3B (green BODIPY fluorescence intensity diagram) in the concerned legend and figures ended to C. Therefore, it was (and is) LDH that belongs to D. Thanks once again.
We take this opportunity of revision to extend our sincere thanks for the meticulous effort put by the learned reviewers resulting in great improvement of the manuscript under consideration.